# Mitochondrial DNA as a Candidate Marker of Multiple Organ Failure after Cardiac Surgery

**DOI:** 10.3390/ijms232314748

**Published:** 2022-11-25

**Authors:** Evgeny Grigoriev, Anastasia V. Ponasenko, Anna V. Sinitskaya, Artem A. Ivkin, Roman A. Kornelyuk

**Affiliations:** Research Institute for Complex Issues of Cardiovascular Diseases, 650002 Kemerovo, Russia

**Keywords:** systemic inflammatory response, cardiac surgery, extracorporeal circulation, mitochondrial DNA, multiple organ failure

## Abstract

Assess the level of mitochondrial DNA depending on the presence of multiple organ failure in patients after heart surgery. The study included 60 patients who underwent surgical treatment of valvular heart disease using cardiopulmonary bypass. Uncomplicated patients were included in the 1st group (*n* = 30), patients with complications and multiple organ failure (MOF) were included in the 2nd group (*n* = 30). Serum mtDNA levels were determined by quantitative real-time polymerase chain reaction with fluorescent dyes. Mitochondrial DNA gene expression did not differ between group before surgery. Immediately after the intervention, cytochrome B gene expression was higher in the group with MOF, and it remained high during entire follow-up period. A similar trend was observed in cytochrome oxidase gene expression. Increased NADH levels of gene expressions during the first postoperative day were noted in both groups, the expression showed tendency to increase on the third postoperative day. mtDNA gene expression in the “MOF present” group remained at a higher level compared with the group without complications. A positive correlation was reveled between the severity of MOF according to SOFA score and the level of mtDNA (r = 0.45; *p* = 0.028) for the end-point “First day”. The ROC analysis showed that mtDNA circulating in plasma (AUC = 0.605) can be a predictor of MOF development. The level of mtDNA significantly increases in case of MOF, irrespective of its cause. (2) The expression of mtDNA genes correlates with the level of MOF severity on the SOFA score.

## 1. Introduction

Given the ineffectiveness of conservative therapy for coronary heart disease valvular heart pathologies and a number of other cardiac diseases, heart surgery under cardiopulmonary bypass has become the gold standard of treatment [1]. Surgical procedures performed using cardiopulmonary bypass may lead to the development of a systemic inflammatory reaction, which is associated with the implementation of the danger theory or danger model. Systemic inflammatory reactions can occur because of the number of factors: cardioplegic arrest, ischemia and reperfusion, contact of blood with nonbiological surfaces, hemodilution, blood transfusion, inhalational and non-inhalational anesthesia [2,3,4]. During the last decade, an increasing number of researchers have started paying attention to mitochondria as cellular organelles responsible for the energy supply of the cell, protein synthesis and programmed cell death, and to the participation of mitochondrial components in the implementation and activation of the signals within the danger model in critical condition [5,6]. The human mtDNA (mitochondrial deoxyribonucleic acid) includes 165,969 base pairs, and contains 2 ribosomal RNAs (ribonucleic acid), 22 transfer RNAs, and 13 protein coding genes that determine the enzymes that are involved in the process of respiration, mtDNA replication, and one non-coding region called the D-loop, which controls the processes of transcription and translation. The human mtDNA is released from the cell under the influence of stress and other events associated with severe conditions [7]. Mitochondria contain several copies of mtDNA, the number of which is interrelated with the size and number of mitochondria, which vary depending on the cell’s energy needs, oxidative stress and various pathological conditions. The number of mtDNA copies reflects the functional state of mitochondria through ATP (adenosine triphosphate) production and enzyme activity [8]. Moreover, mtDNA can be damaged by reactive oxygen species (ROS), which leads to mitochondrial dysfunction, systemic inflammation and activation of programmed cell death [9]. High levels of circulating mtDNA act like DAMPs (danger associated molecular patterns), inducing inflammation and organ damage [10]. Over the past few years, the association between the level of circulating mtDNA and various pathologies has been actively studied. Thus, the authors of one study note an increase in circulating mtDNA and cytokines in plasma in all elderly patients without assessing the severity of the condition and the postoperative status after minimally invasive aortic valve replacement [11]. Another group of researchers found that patients with cardioplegic arrest had higher expression of COX III (cyclooxygenase III), NADH1 (Nicotinamide adenine dinucleotide 1), NADH2, and Cyto B (cytochrome B) genes compared to healthy participants. However, decreased expression of COX III, NADH1, NADH2 genes was observed after implementing the method of targeted temperature management of 33C [12]. Chinese scientists have shown that the level of circulating mtDNA is higher in trauma patients compared to healthy patients. It is worth noting that the high plasma levels of mtDNA persisted in patients with post-traumatic inflammatory reactions [13,14].

The aim of this study was to assess the level of serum mtDNA in patients who underwent surgery for valvular heart disease, depending on the development of postoperative MOF (multiorgan failure).

## 2. Results

### 2.1. MtDNA Levels in Patients

In both groups, gene expression level of mtDNA was comparable at baseline and did not differ between the groups at the preoperative stage. At the postoperative stage, cytochrome B gene expression was significantly higher in the group with MOF and it remained high throughout follow-up period; there were no changes in gene expression in the group without MOF (Figure 1). A similar trend was observed with cytochrome oxidase gene expression (Figure 2 and Figure 3). Increased NADH levels during the first postoperative day was noted in both groups, and the expression showed a tendency to increase on the third postoperative day (Figure 4 and Figure 5). Gene expression level sof mtDNA remained at a higher level in the “MOF present” group or MOF + in figures compared to the group without complications («MOF −»).

Next, we performed a regression analysis between the indicators of the severity of MOF according to SOFA score and the level of serum mtDNA in patients at the preoperative stage and on the first, third and seventh days after surgery in the group with complications. A positive correlation was revealed (r = 0.45; *p* = 0.028) for the “First day” endpoint (Figure 6).

In order to assess mtDNA as a predictor of the development of postoperative complications, ROC analysis was performed, showing that the level of circulating mtDNA in serum (AUC = 0.605) can be a predictor of the development of MOF (Figure 7).

### 2.2. Features of MOF in Patients with Complications

Analyzing the features of postoperative multiple organ failure, a clear prevalence of acute kidney injury among the manifestations of MOF (more than 80% vs. ARDS 38%, acute intestinal distress syndrome 13%, and acute cerebrovascular insufficiency in the form of depression and/or delirium 40%) was established (Figure 8).

## 3. Discussion

We have noted a significant increase in the level of circulating mtDNA in selected genes, primarily on the first day after admission to the ICU (intensive care unit), along with a number of clinical factors that can predict the development of MOF in critically ill patients with MOF after heart surgery performed using cardiopulmonary bypass, which was mentioned in our previous publication [15]. Performing surgeries using a cardiopulmonary bypass is associated with the development of a systemic inflammatory reaction due to ischemia-reperfusion injury (primarily in the myocardium), contact of blood with nonbiological surfaces and the release of endotoxins. SIRS (systemic inflammatory response) is characterized by a “cytokine release”; in case of maladaptation, it can lead to irreversible organ dysfunction [16]. SIRS, initiated by DAMP molecules, are of non-infectious type [17]. DAMPs are able to activate epithelial, endothelial cells and fibroblasts, as well as neutrophils, macrophages and dendritic cells [18]. Activation of these cells promotes the release of cytokines and chemokines into the bloodstream, which in turn leads to the development of inflammatory reactions and activation of the immune response [19]. Currently, extracellular mtDNA acting as a DAMP molecule is of great interest to many researchers [20,21]. The release of mtDNA is associated with an increase in reactive oxygen species and a decrease in membrane permeability. There are 2 mechanisms described in the literature that indicate that mtDNA contributes to the development of inflammatory reactions. The first one is associated with the activation of NLRP3 (Nucleotide-binding oligomerization domain) inflammasome, which in turn leads to the activation of caspase-1, and as a result, pro-inflammatory cytokines (IL1β, IL18) become activated. The second mechanism is based on the activation of TLR9, which binds to unmethylated CpG motifs in bacterial and viral DNA [22]. Both mechanisms ultimately lead to the same result–SIRS. Our study confirms the hypothesis that mtDNA can act as a potential initiator and biological marker for predicting the development of SIRS in patients undergoing cardiac surgery. Earlier studies conducted by scientists from different countries prove that elevated mtDNA levels can be observed in various conditions, such as breast cancer, ischemic stroke, myocardial infarction, and systemic inflammatory response [22]. For example, a team of Chinese researchers showed that the level of serum mtDNA in patients with trauma is higher compared to healthy donors. Moreover, the level of serum mtDNA was also higher in the group of patients who developed post-traumatic SIRS [23]. In our study, there is a slight difference (1.25-fold) in the level of mtDNA in the group of patients with SIRS, which can be explained by the small sample. In other studies [23], there is a positive correlation between the SOFA score (Sequential Organ Failure Scale) and the level of serum mtDNA in patients on the 1st postoperative day, which is reflected in our study as well. A group of researchers has shown that the level of serum mtDNA in patients allows specialists to predict the survival rate of patients in the ICU (Intensive Care Unit). Thus, patients who died on the 28th day after admission had higher mtDNA copy number compared to the surviving patients [24].

The association between the gene expression level of mtDNA and the severity of MOF has been discussed in previous studies primarily devoted to sepsis and trauma [25,26]. Scientists have noted that SIRS, as a manifestation of a critical condition, is accompanied by the development of AKI (Acute Kidney Injury) in at least 35% of cases, which may indicate mtDNA’s activity as an alarmin even in the absence of a primary focus of infection and in case of non-infectious SIRS. This is exactly what can be observed in cardiac surgery patients with complications. The authors emphasize that the level of serum mtDNA does not possess such great diagnostic value for AKI compared to the level of mtDNA in urine. 

Similarly to the above-mentioned authors, we have obtained results which indicate an association between an increase in mtDNA levels and the development of MOF and an unfavorable outcome (we did not evaluate the association between mtDNA levels and the outcome). The authors note that mtDNA surpassed the diagnostic significance of such “classical” prognostic parameters of MOF as blood transfusion volume, tissue damage volume (in case of trauma as the cause of critical condition and MOF) and the severity of injury. Similarly, the authors established a direct association between the severity of SIRS and the level of mtDNA (a strong linear relationship). We have also obtained comparable data, in that in case of unfavorable course of MOF and its persistence, the level of mtDNA remains consistently high, which may be due to the immune suppression and mtDNA’s contribution to it (previously, the role and contribution of immunosuppression was analyzed and demonstrated in our studies on the patient-specific cardiovascular model) [27]. The association between the level of mtDNA in patients with trauma and sepsis (in the presence of non-infectious and infectious SIRS) and the manifestation of MOF may confirm the theory of the development of distant organ damage that closes the chain of pathophysiological events involving MOF in case of damage to one organ. These results allow us to conclude that mtDNA can realize its prognostic role as an alarmin and candidate prognostic marker in the first 8 h [28].

Our study has a number of limitations. The study is single-center, which means only small number of patients can be included due to the scheduling of surgical procedures. Further research involving larger sample and the use of mtDNA as a marker for assessing the effectiveness of interventions is necessary. Moreover, the analysis of methods of extracorporeal blood purification based on the patient’s endophenotype or the use of antioxidants (previously demonstrated on models of trauma, burn injury and sepsis) from the point of view of the most effective therapeutic method and the most optimal method of blood purification (for example, cytokine adsorption, capable of adsorbing alarmins) using measurement of alarmins is necessary as well [29,30]. For us, it was fundamental to standardize the methods of perfusion and intensive treatment within each of the groups in order to exclude the influence of a particular type of therapy on the course of MOF and to exclude the effect of therapy on the level of the studied parameters of mtDNA. However, it is impossible to completely exclude the impact of such methods as extracorporeal blood purification on the level of the studied biomarkers, which requires further subgroups within the group with a complicated course in terms of assessing the impact of individual methods (blood purification, nutritional support, etc.).

## 4. Material and Methods

### 4.1. Cohort Description

The study included patients who received surgical treatment in the period from 2018 to 2020. All patients were informed about the purpose of the study and provided informed consent before participation. The study protocol received approval by the Ethics Committee of the Research Institute. Detailed clinical characteristics of patients are presented in Table 1.

All patients were treated using a nonpulsatile cardiopulmonary bypass (normothermic continuous perfusion) at flow rate of 2.3 L/min/m^2^. Crystalloid cardioplegia with Kustadiol solution (DrKohler Chemi, Bensheim, Germany) was used for myocardial protection. Depending on the severity of myocardial hypertrophy, cardioplegia was delivered antegradely or retrogradely. Surgeries were performed under endotracheal anesthesia, starting with propofol (3 mg/kg/h), then anesthesia was maintained with sevoflurane (0.8–1.3 MAC) before the operative stage, following that a dose of 2 mg/kg/h propofol was used at the operative stage, then after the removal of aortic cross-clamping, sevoflurane administration was resumed. Analgesia was performed with fentanyl at a dose of 5–7 μg/kg/h. Intraoperative hemodynamic monitoring included: invasive blood pressure monitoring, cardiac output monitoring (either using a Swan-Ganz catheter or intraoperative transesophageal echocardiography), and Bispectral Index monitoring. The severity of multiple organ failure in the postoperative period was determined using the SOFA scoring system. The following criteria were used to describe the progression of organ failure: ARDS (acute respiratory distress syndrome) according to Berlin definition, AKI according to ADQI (Acute Dialysis Quality Initiative) definitions, CNS (Central Nervous System) according to CAM-ICU (Confusion Assessment Method-Intensive Care Unit) delirium severity scale) [12]. 

### 4.2. Blood Sampling

Venous blood was collected from patients in test tubes containing K3-EDTA (Ethylenediaminetetraacetic acid) and a clot activator at the following stages: preoperatively, on the 1st, 3rd and 7th postoperative day. Next, the blood was centrifuged, and the serum was aliquoted into Eppendorf tubes and stored at −80 °C for the study.

### 4.3. DNA Isolation and Quantitative mtDNA Analysis

DNA was isolated from 200 µL of serum using the QIAamp DNA Mini Kit (Cat. no 51306, Qiagen, Hilden, Germany), concentration was calculated on a NanoDrop 2000 Spectophotometer (Thermofisher Scientific, Waltham, MA, USA), and the DNA was stored at −80 °C. mtDNA gene expression was determined by quantitative PCR (polymerase chain reaction) with the SYBR Green dye on a CFX96 Touch Real-Time PCR Detection System (Bio-Rad, Hercules, CA, USA). The selection of primers was carried out based on the analysis of the literature data (NADH, NADH 1, NADH 2, COX II, COX III, Cytochrome b) [13,31]. The primers were synthesized by “Eurogen” CJSC (Moscow, Russia). The final volume of the reaction mixture of 10 mL contained 5 mL of the PowerUp SYBR Green Master Mix (Applied biosystems, Bedford, MA, USA), 500 µL of forward and reverse primers and 20 µL of DNA. PCR was performed in standard 96-well plates containing 26 analyzed samples, five standards with 2-fold dilution and one negative control. Amplification included 40 cycles consisting of denaturation at 95 °C for 15 s, annealing at 52–56 °C for 15 s (depending on the melting temperature of the primers), and extension at 72 °C for 1 min. Normalization of PCR data was carried out using GAPDH (Glyceraldehyde 3-phosphate dehydrogenase) reference gene (housekeeping gene). The relative expression ratio was calculated by ΔCt method. Charts showing mtDNA gene expression are presented using conventional units (gene expression level normalized to the reference gene).

### 4.4. Statistical Data Processing

Statistical data processing was carried out using the GraphPad Prism 8.0 software. Data are presented as median, 25th and 75th percentiles. Normality of data distribution was tested by using the Kolmogorov-Smirnov test. The Mann-Whitney U-test (for pairwise comparison) and the Kraskell-Wallis test (for comparing several groups) were used for intergroup comparison. ROC analysis was used to assess the prognostic value of the predictor, the AUC was used to evaluate the overall prognostic performance. The correlation between the nonparametric values was calculated using linear regression.

## 5. Conclusions

The level of mtDNA significantly increases in case of the development of MOF, regardless of its cause, in a small cohort of cardiac surgery patients.The level of mtDNA correlates with the severity of MOF on the SOFA score.

## Figures and Tables

**Figure 1 ijms-23-14748-f001:**
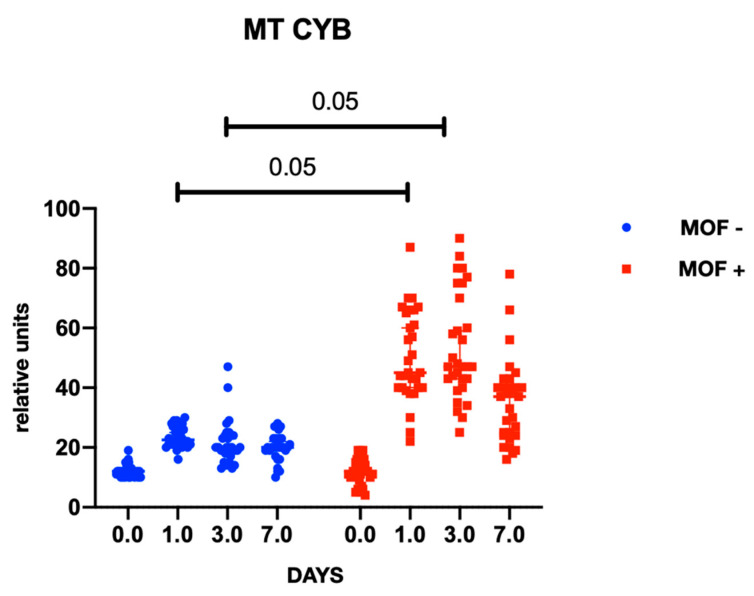
Cytochrome B gene expression.

**Figure 2 ijms-23-14748-f002:**
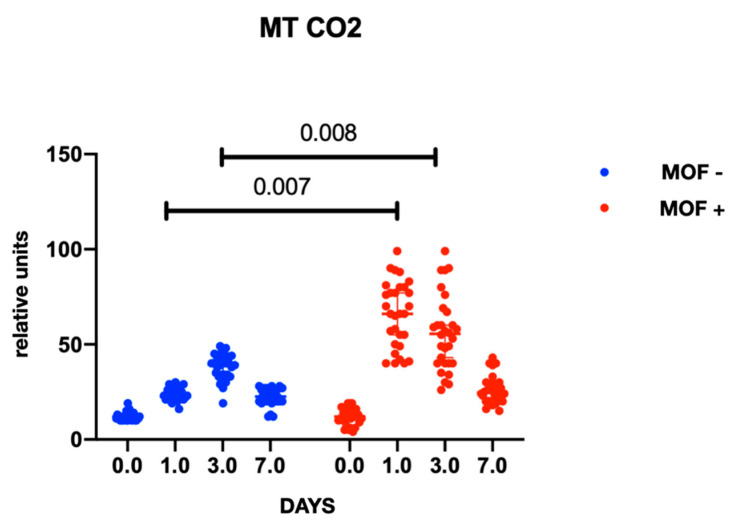
Cytochrome oxidase 2 gene expression.

**Figure 3 ijms-23-14748-f003:**
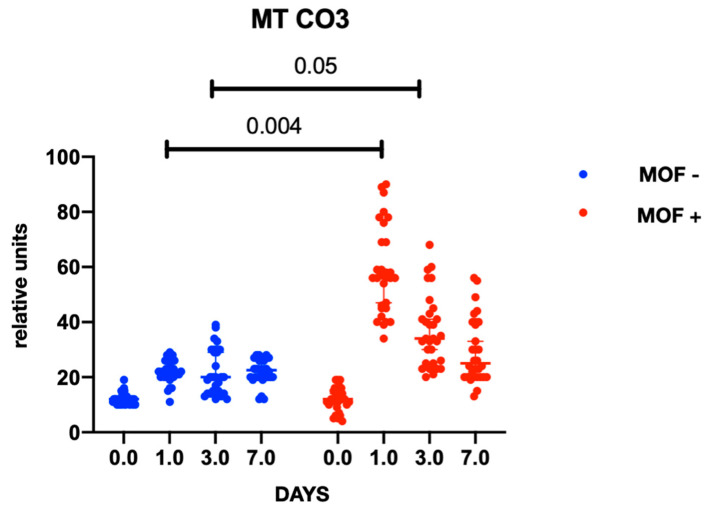
Cytochrome oxidase 3 gene expression.

**Figure 4 ijms-23-14748-f004:**
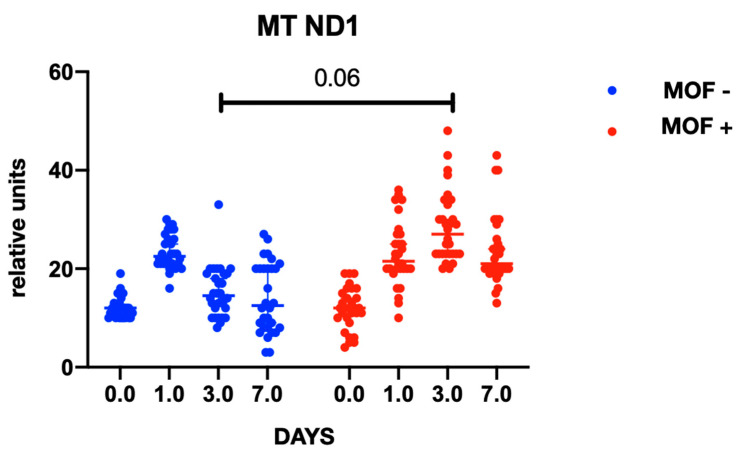
NADH1 gene expression.

**Figure 5 ijms-23-14748-f005:**
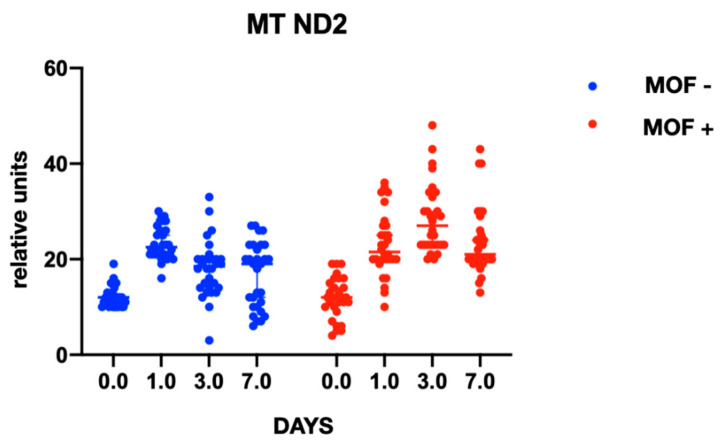
NADH2 gene expression.

**Figure 6 ijms-23-14748-f006:**
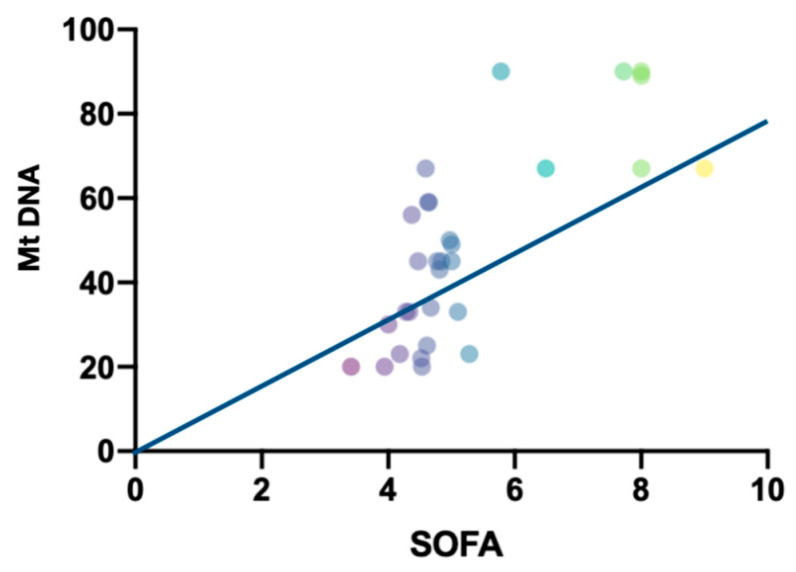
Regression analysis between the indicators of the severity of MOF according to SOFA score and the level of plasma mtDNA in patients (r = 0.45 and *p* = 0.028).

**Figure 7 ijms-23-14748-f007:**
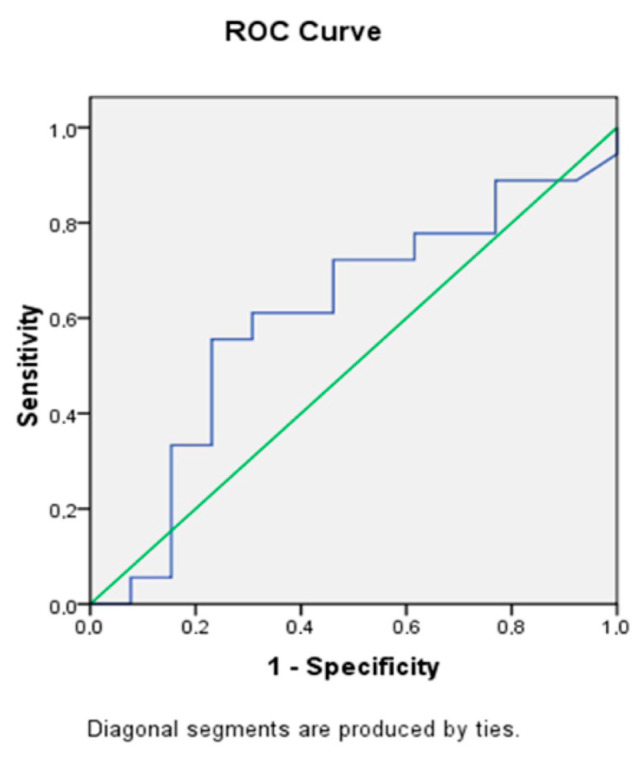
ROC curve.

**Figure 8 ijms-23-14748-f008:**
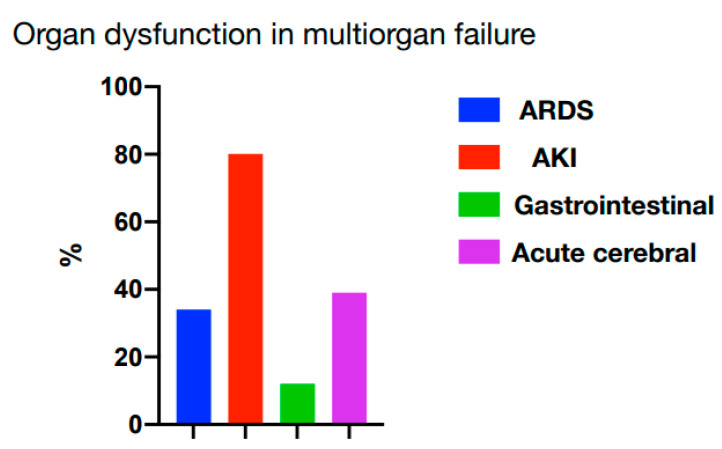
Organ dysfunction in multiorgan failure structure (ARDS—acute respiratory distress syndrome, AKI—acute kidney injury).

**Table 1 ijms-23-14748-t001:** Clinical characteristics of patients.

Parameter	Patients without Complications (Group 1, *n* = 30)	Patients with Complications (Group 2, *n* = 30)	*p*
Age, years old	57.2 (42.0–66.0)	59.2 (48.0–66.9)	ns
Male, abs. (%)	15 (50)	15 (41.6)	ns
Body weight index, kg/m^2^	33.4 (32.1–35.2)	36.4 (32.0–36.7)	ns
Comorbidity by CIRS, points	9.6 (8.0–11.1)	9.0 (8.9–11.5)	ns
Aortic valve surgery, abs (%)	5 (16.6)	7 (23)	ns
Mitral valve surgery, abs aбc. (%)	6 (20)	3 (10)	ns
Aortic + mitral valve surgery, abs (%)	19 (63.4)	20 (67)	ns
The reasons of MOF, abs (%):			
- low cardiac output syndrome, shock - acute blood loss, shock - combination	Not applicable	11 (36) 7 (23) 12 (41)	0.0001
The time of artificial circulation min	121 (89–167)	158 (101–201)	0.001
Aortic cross clump, min	98 (90–122)	108 (99–145)	ns
The need for transfusion after surgery, %	0	100%	

Note: CIRS—Cumulative Illness Rating Scale. ns–nonsignificant.

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
