# Peer review of "Mitochondrial DNA as a Candidate Marker of Multiple Organ Failure after Cardiac Surgery"

_ijms, 2022, doi:10.3390/ijms232314748_

Round 1
Reviewer 1 Report
Grigoriev and coworkers analyzed serum samples from surgical patients (valvular heart disease) with and without complications regarding mtDNA gene expression. Some genes are identified whose expression is significantly different.
Comments:
- Line 2: Dna should be DNA
- Line 8: Why is this investigated? A reason for the study should be mentioned.
- Line 12: Sometimes serum is mentioned as the starting material, sometimes plasma...
- Line 16: From the results, it is clear that not the coenzyme NADH is meant, but the mRNA expression of NAPH1 and NADH2. This is confusing and needs revision.
- A list of abbreviations is shown at the end. In the text, however, sometimes abbreviations are explained when first mentioned, sometimes not... this should be consistent! E.g. in the abstract: MOF is explained, NADP, SOFA, ROC not...
- line 18: genes expression should be gene expression
- Line 29/30: By-pass surgery may be the surgical gold standard in the treatment of CHD/CVD, but generally, internal medicine procedures are now more common.
- Line 52: Explain DAMPs
- Figures: The images are much too large. What is indicated: Mean, median, SD, SEM? With the selected representation, however, they are could be hardly recognized...The axis labeling is not complete and is often in Russian. The same applies to the designation of the groups. There is also a complete legend missing for each figure, from which it is clear what exactly was done.
- Figure 6: The regression line is missing, as well as the information on r and p in the figure.
- Line 99-101: The ROC analysis should be shown.
- Line 110: Several times in the text, the level of circulating mtDNA is mentioned. This is not true since only selected genes were analyzed.
- Line 129: IL-1b should be IL-1β
- Line 130: TLR-9 should be TLR9
- Table 1: In the table is also again a small part in Russian. Wherever there are commas, there must be dots. The first two rows must be formatted differently so that everything fits on one level. The information about the p-values is strange.... 0.1, what is meant by this? Not significant? Either n.s. or the exact p-value should be used
- Line 224-226: 95°C, 52-55°C, 72°C
Author Response
Dear colleagues, thank you very much for reviewing the article. We respond to comments. The modified article will be uploaded to the editorial system.
Comments:
- Line 2: Dna should be DNA. Correct
- Line 8: Why is this investigated? A reason for the study should be mentioned. Correct
- Line 12: Sometimes serum is mentioned as the starting material, sometimes plasma... Correct in line 213 and 214
- Line 16: From the results, it is clear that not the coenzyme NADH is meant, but the mRNA expression of NAPH1 and NADH2. This is confusing and needs revision. Correct
- A list of abbreviations is shown at the end. In the text, however, sometimes abbreviations are explained when first mentioned, sometimes not... this should be consistent! E.g. in the abstract: MOF is explained, NADP, SOFA, ROC not... Correct all the abbreviations in text
- line 18: genes expression should be gene expression Correct
- Line 29/30: By-pass surgery may be the surgical gold standard in the treatment of CHD/CVD, but generally, internal medicine procedures are now more common. Correct
- Line 52: Explain DAMPs Correct
- Figures: The images are much too large. What is indicated: Mean, median, SD, SEM? With the selected representation, however, they could be hardly recognized...The axis labeling is not complete and is often in Russian. The same applies to the designation of the groups. There is also a complete legend missing for each figure, from which it is clear what exactly was done. Correct all figures
- Figure 6: The regression line is missing, as well as the information on r and p in the figure. Correct
- Line 99-101: The ROC analysis should be shown. Correct
- Line 110: Several times in the text, the level of circulating mtDNA is mentioned. This is not true since only selected genes were analyzed. Correct
- Line 129: IL-1b should be IL-1β Correct
- Line 130: TLR-9 should be TLR9 Correct
- Table 1: In the table is also again a small part in Russian. Wherever there are commas, there must be dots. The first two rows must be formatted differently so that everything fits on one level. The information about the p-values is strange.... 0.1, what is meant by this? Not significant? Either n.s. or the exact p-value should be used Correct
- Line 224-226: 95°C, 52-55°C, 72°C Correct
Reviewer 2 Report
"The number of mtDNA copies reflects the functional state of mitochondria through ATP production and enzyme activity" Please explain it deeper cause is involved in the results achieved here.
"In both groups gene expression level of mtDNA was comparable at baseline and did not differ between ..." and between patients and general population?; please cite here.
"we did not evaluate the association between mtDNA levels and the outcome" L-157 Why? It could be the most interesting part of the MS, conclusions found here are quite similar to those observed in cite 25 for ICU.
treatment has been taken into consideration? If so it should be included in methodology.
Author Response
Dear colleagues,
Thank you very much for reviewing the article. We respond to comments. The modified article will be uploaded to the editorial system.
Answers:
"The number of mtDNA copies reflects the functional state of mitochondria through ATP production and enzyme activity" Please explain it deeper cause is involved in the results achieved here. There is an assumption that the activity of mitochondria and the activity of biosynthesis can affect the level of expression of mt DNA, which links both the work of mitochondria, as well as the level of free mt DNA in the blood serum, which can be combined by the presence of an inflammatory-metabolic complex.
2. "In both groups gene expression level of mtDNA was comparable at baseline and did not differ between ..." and between patients and general population?; please cite here. The level of basic gene expression did not differ, which indicated the comparability of patients at the start of the study.
3. "We did not evaluate the association between mtDNA levels and the outcome" L-157 Why? It could be the most interesting part of the MS, conclusions found here are quite similar to those observed in cite 25 for ICU. We can assume that there is a correlation between the level of the studied biomarkers and the level of mortality as such in critical patients, however, in the article we present data only in terms of correlation with MOF, since such a serious conclusion as the effect on mortality requires the recruitment of more patients.
4. Treatment has been taken into consideration? If so it should be included in methodology. Lines 201-205. For us, it was fundamental to standardize the methods of perfusion and intensive treatment within each of the groups in order to exclude the influence of a particular type of therapy on the course of MOF and to exclude the effect of therapy on the level of the studied parameters of mt DNA. However, it is impossible to completely exclude the impact of such methods as extracorporeal blood purification on the level of the studied biomarkers, which requires further subgroups within the group with a complicated course in terms of assessing the impact of individual methods (blood purification, nutritional support, etc.).
Round 2
Reviewer 1 Report
The authors made many of the suggested changes, but overlooked or misunderstood some things. Unfortunately, still not all points have been changed accordingly....
Comments:
- Line 13: As before, sometimes serum, sometimes plasma is written in the manuscript (for the own analyses)....
- Line 17: The term NADH levels is misleading! It is NADH expression level. In addition, still not every abbreviation is explained… e.g. NADH.
- Line 22: The authors performed…. This wording is quite uncommon presenting the own data.
- Line 19: To proper term is gene expression!!
- Figures: The display of the Figures is still not optimal. Again the question: What is shown by the lines in the data points? Mean, median? SD or SEM? This information is not found in the figures, nor in the method section. Moreover, it is not recognizable because it is in the same color as the points. That should be in black and shown in the foreground. The labeling of the figures is still very poor. One should be able to tell from this what was made. Investigated samples, used methods… At least like in Fig. 6!
- Line 113: Fig X???
- Line 145: Actually IL-1β and not IL-1beta….
- Table 1: Still need to replace some commas with periods....
- Line 254/55: It is called, e.g. 95°C and not 95 ° C....
Author Response
Dear colleagues, thank you for your request for our manuscript.
- Line 13: As before, sometimes serum, sometimes plasma is written in the manuscript (for the own analyses).... We have tried to correct all missing dates (lines ## 13, 96, 104, 145, 146, 150, 152,
- Line 17: The term NADH levels is misleading! It is NADH expression level. In addition, still not every abbreviation is explained… e.g. NADH. Correct
- Line 22: The authors performed…. This wording is quite uncommon presenting the own data. Correct, we include the ROC curve into text.
- Line 19: To proper term is gene expression!!Correct
- Figures: The display of the Figures is still not optimal. Again the question: What is shown by the lines in the data points? Mean, median? SD or SEM? This information is not found in the figures, nor in the method section. Moreover, it is not recognizable because it is in the same color as the points. That should be in black and shown in the foreground. The labeling of the figures is still very poor. One should be able to tell from this what was made. Investigated samples, used methods… At least like in Fig. 6! We correct dates of statistical analyses in lines ## 253 and 254 and we explain the group settings in lines 81 82 83
- Line 113: Fig X??? We correct numbers of figures
- Line 145: Actually IL-1β and not IL-1beta…. Correct
- Table 1: Still need to replace some commas with periods.... Correct
- Line 254/55: It is called, e.g. 95°C and not 95 ° C....Correct